# Democratizing Clinical Risk Prediction with Cross-Cohort Cross-Modal Knowledge Transfer

**Qiannan Zhang, Manqi Zhou, Zilong Bai, Chang Su, Fei Wang**[*]
Weill Cornell Medicine, Cornell University
{qiz4005,maz4010,zib4001,chs4001,few2001}@med.cornell.edu

## Abstract

Clinical risk prediction plays a crucial role in early disease detection and personalized intervention. While recent models increasingly incorporate multimodal data, their development typically assumes access to large-scale, multimodal datasets and substantial computational resources. In practice, however, most clinical sites operate under resource constraints, with access limited to EHR data alone and insufficient capacity to train complicated models. This gap highlights the urgent need to democratize clinical risk prediction by enabling effective deployment in data- and resource-limited local clinical settings. In this work, we propose a cross-cohort cross-modal knowledge transfer framework that leverages the multimodal model trained on a nationwide cohort and adapts it to local cohorts with only EHR data. We focus on EHR and genetic data as representative multimodal inputs and address two key challenges. First, to mitigate the influence of noisy or less informative biological signals, we propose a novel mixture-of-aggregations design to enhance the modeling of informative and relevant genetic features. Second, to support rapid model adaptation in low-resource sites, we develop a lightweight graph-guided fine-tuning method that adapts pretrained phenotypical EHR representations to local cohorts using limited patient data. Extensive experiments on real-world clinical data validate the effectiveness of our proposed model.

## 1 Introduction

Predictive diagnosis and risk prediction play a pivotal role in clinical practice, particularly for chronic conditions such as Alzheimer's disease and related dementias, which stem from a complex interplay of genetic and environmental factors and impose significant socioeconomic burdens and public health challenges. As current treatments may slow progression yet cannot reverse the pathological process, early risk prediction before clinical diagnosis aims for personalized monitoring, proactive management and preventive care in advance to slow disease progression and improve health outcomes.

Recent studies on disease risk prediction primarily rely on electronic health record (EHR) data to capture real-world patient phenotypical information [21, 27, 50, 43], including diagnoses, prescriptions, etc. A growing number of research has explored introducing additional biological modalities, such as medical imaging, genomics and proteomics, to boost predictive performance [22, 40, 10, 2, 56, 59]. However, most existing approaches are developed for a single cohort, under the assumption that large patient populations, modalities, and ample computational resources are available for training. *In reality, this assumption often fails to hold, as many regional health providers may contend with insufficiency of patient samples, costly modalities, and computing infrastructures to train predictive models locally.* Thus, democratizing clinical risk prediction capability to local cohorts is an important step toward enabling accessible and equitable healthcare solutions across diverse clinical scenarios.

---

[*]Corresponding Author

39th Conference on Neural Information Processing Systems (NeurIPS 2025).

Inspired by the growing development of nationwide biobanks such as the All of Us Research Program and UK Biobank [6, 35], which collect high-volume multimodal data besides EHRs to advance biomedical research, and driven by the goal to democratize predictive modeling for local clinical sites constrained to EHRs alone, this study investigates the following research question: *Can we develop a predictive model using multimodal data from a nationwide cohort and transfer it to resource-limited local cohorts with only EHR data, enabling accurate risk prediction without sharing sensitive patient information?*

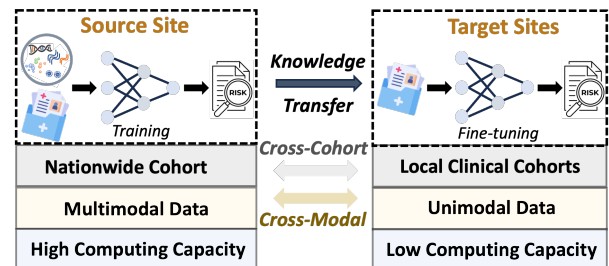

Figure 1: Conceptual diagram of cross-cohort cross-modal knowledge transfer.

Despite its appeal, developing such a model is non-trivial due to *two challenges*. *First*, multimodal data is inherently noisy or imprecise due to measurement errors and multifactorial biological processes, which can complicate the correlation between multimodal features and disease outcomes, degrading predictive accuracies. *Second*, cross-modal learning from nationwide multimodal cohorts to local unimodal cohorts inevitably faces distribution shifts due to different population characteristics, data collection protocols, or healthcare practices. These shifts necessitate fine-tuning of the transferred model to adapt to the target cohort. *However, the limited data and computational resources at local cohorts make it difficult to perform effective fine-tuning.* Insufficient data increases the risk of overfitting, while constrained computational capacity limits the use of powerful models that typically require substantial computational overhead.

Motivated by these challenges, this paper proposes a cross-cohort, cross-modal knowledge transfer framework named $\mathbf{C}^3\mathbf{M}$ to support the democratization of clinical risk prediction at resource-limited local clinical sites. In this work, we focus on utilizing EHR and genetic modalities on the source biobank due to the wide availability of EHRs across cohorts and the essential information that genetic data offers to capture individual-level biological mechanisms associated with disease risk [15]. To address the *first challenge* of noisy and imprecise genetic data, we design a strategy for gene encoding with mixture-of-aggregations. This approach employs multiple aggregation tokens to jointly identify and aggregate meaningful genetic features from different views through attention modulation and feature reconstruction, while mitigating the influence of noisy or less informative signals. To address the *second challenge* involving limited data and computational resources that hinder model fine-tuning, we draw inspiration from the increasing use of foundation models. Specifically, we adopt an EHR foundation model as the EHR encoder, which enables rapid and efficient phenotypical EHR representation generation, even when local cohorts have only a small number of patients. However, fine-tuning a foundation model remains extremely difficult in resource-constrained clinical sites due to the substantial computational demands. To overcome this limitation, we propose a lightweight fine-tuning method that utilizes limited patient data by constructing a bipartite graph between patients and the medical concepts recorded in their EHRs. A graph neural network (GNN) is then applied to this bipartite graph to adjust the foundation model–initialized phenotypical embeddings based on their associated medical concepts, achieving effective adaptation without modifying the foundation model itself. Finally, with the obtained phenotypical EHR and genetic embeddings and the goal of transferring a model to a resource-limited local cohort with unimodal data, we leverage teacher-student distillation to achieve cross-modal knowledge transfer. The teacher prediction model is distilled into a student prediction model, which is then transferred along with the GNN to the local cohort for further fine-tuning and downstream risk prediction.

Our contributions can be summarized as follows: 1) We study the crucial and practical problem of democratizing clinical risk prediction capabilities in resource-limited local cohorts by leveraging knowledge from nationwide multimodal cohorts; 2) We propose a novel mixture-of-aggregations design to enhance the modeling of informative and relevant genetic features while mitigating the influence of noisy or less informative signals; 3) We develop a lightweight graph-guided fine-tuning method that adapts pretrained phenotypical EHR representations to target cohorts using limited patient data, without requiring updates to the foundation model; 4) We conduct extensive experiments on real-world clinical data to demonstrate the effectiveness and generalizability of our model.

## 2 Related Work

**Cross-modal learning.** Conventional multimodal learning methods aim to capture complementary information across modalities by multimodal fusion [3, 59], typically assuming complete modality availability. However, modality missingness, common in practice, presents a widespread challenge to their applicability [54]. Therefore, research has focused on learning cross-modal knowledge to compensate for missing information, falling in two main directions. One line of work relies on modality completion that reconstructs the missed modalities from observed ones through generative networks [44, 68, 7, 25, 16, 51, 19, 34, 53, 29]. In contrast, modality-robust learning seeks to build representations that remain effective under incomplete observations without explicit reconstruction. Common strategies explore adaptive integration [28, 57], invariant representations [47, 52, 61] or graph-based relations [62, 58, 63, 64, 55] to leverage the modality interactions. Besides, knowledge distillation is also employed where the teacher model with multimodal data supervises modality-incomplete students to transfer cross-modal knowledge [36, 37, 49, 20, 48]. These methods are primarily designed to handle random or block-wise missingness, under assumptions of modality availability at test time, distributional consistency across modalities, or in-domain training and deployment. They often struggle to generalize when certain modalities are consistently absent at deployment, and become less applicable in our setting, where modality availability and data distribution differ across cohorts.

**Cross-cohort knowledge transfer.** The healthcare community increasingly advocates for cross-cohort model deployment to facilitate the transfer of beneficial knowledge, drawing inspiration from cross-domain transfer learning across disciplines [24, 66, 60, 65]. Despite its promise to enhance predictive diagnosis and risk prediction through the use of multi-institutional data, the question of how to achieve effective multimodal learning across cohorts remains largely open and underexplored due to practical constraints on data sharing and computational resources [38, 4, 18]. In fields such as computer vision and natural language processing, multimodal cross-domain adaptation addresses domain shifts to facilitate knowledge transfer with multiple modalities. Classical approaches primarily focus on reducing domain discrepancies by aligning representations through domain adversarial learning or by extracting domain-invariant features [23, 26, 30, 9, 11, 13]. These methods typically assume consistent modality availability, shared samples across domains, and simultaneous access to both source and target domains during training. A few works consider more realistic scenarios, where the target domains lack certain modalities or have non-overlapping modalities compared to the source [67], while they still necessitate joint training with target domains. In contrast, our setting assumes no access to the target domain during training and involves consistent modality missingness at deployment across all target samples. This poses a more challenging generalization problem, requiring source-trained models to exhibit stronger cross-cohort transferability.

## 3 Preliminaries

**Problem Definition.** We consider the task of risk prediction in a cross-cohort cross-modal setting, where the source cohort comprises a nationwide patient population with rich multimodal data, while the target cohort consists of a local patient population with only EHR data available. In this paper, we focus on two modalities: electronic health records (EHRs) and genetic profiles. Specifically, in the source cohort $\mathcal{S} = \{(r_i, g_i, y_i)\}_{i=1}^{N_S}$, where $N_S$ denotes the total number of patients, each patient $i$ is associated with (1) a sequence of longitudinal EHR events $r_i = \{e_i^1, e_i^2, \ldots, e_i^{|r_i|}\}$ extracted from their medical history, (2) a binary genetic profile $g_i \in \{0, 1\}^N$ indicating the absence or presence of genetic mutations for the $N$ genes under consideration, and (3) a disease label $y_i \in \{0, 1\}$ indicating the occurrence of the outcome. EHR events $r_i$ capture the patient's phenotypical information, where each event $e_i$ corresponds to a coded medical concept, e.g., a diagnosis or a prescribed drug. Here we denote the set of medical concepts as $\mathcal{C}$. To enable early prediction, only EHR events occurring prior to the prediction time are utilized. A prediction model $\mathcal{F}(r_i, g_i) \rightarrow \hat{y}_i \in [0, 1]$ is trained on this cohort to estimate the disease risk using both EHR and genetic data. In the target cohort $\mathcal{T} = \{(r_j, y_j)\}_{j=1}^{N_T}$ consisting of $N_T$ patients, only EHR data is available. This reflects a practical scenario in local cohorts, where biological data such as genetic profiles are typically unavailable. The target cohort may differ from the source cohort demographically and clinically.

In this paper, we aim to develop a multimodal prediction model using the source cohort $\mathcal{S}$ that can be effectively transferred and adapted to the unimodal, distribution-shifted target cohort $\mathcal{T}$, under the

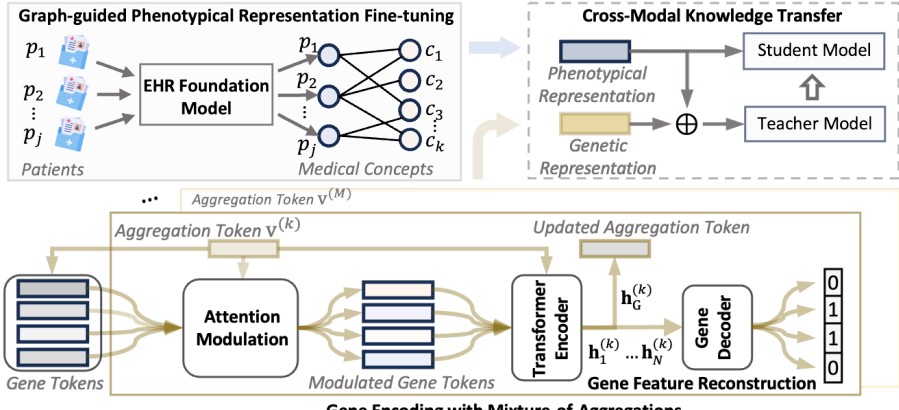

Figure 2: An overview of the $\mathbf{C}^3\mathbf{M}$ framework: 1) Gene encoding with mixture-of-aggregations identifies the most relevant and informative genes to learn genetic representation; 2) Graph-guided phenotypical representation fine-tuning adapts pre-trained EHR embeddings through a lightweight design; 3) Cross-modal knowledge transfer facilitates model transfer to target cohorts for gene-free deployment, by distilling knowledge from both EHR and genetic modalities.

constraint that data from $\mathcal{S}$ and $\mathcal{T}$ cannot be jointly accessed due to privacy-preserving, non-sharing restrictions. To facilitate adaptation, we assume access to a small labeled subset $\mathcal{T}_{\text{labeled}} \subset \mathcal{T}$, which can be used to fine-tune the model using only EHR data available in the target cohort.

**MOTOR.** MOTOR (Many Outcome Time-Oriented Representations) [42] is a pretrained foundation model designed to encode structured EHRs into patient-level embeddings. Given a longitudinal EHR record $r = \{e^1, e^2, \ldots, e^{|r|}\}$, where each event $e^t$ may include clinical codes (e.g., diagnoses, medications, lab tests, etc.) and corresponding timestamps, MOTOR transforms this sequence into a fixed-dimensional embedding vector that captures the temporal dynamics and clinical context of the patient's history. The model is pretrained on large-scale EHRs and claims data using a self-supervised time-to-event prediction objective, thus learning informative representations for survival and risk modeling tasks.

## 4 The Proposed Method

In this section, we present our proposed model, $\mathbf{C}^3\mathbf{M}$, which comprises three key components: 1) gene encoding with mixture-of-aggregations, 2) graph-guided EHR representation fine-tuning, and 3) cross-modal knowledge transfer with distillation. The overall framework is illustrated in Figure 2.

### 4.1 Gene Encoding with Mixture-of-Aggregations

In the source cohort, for patients with available genetic profiles, we first aim to encode the genetic data into representations. With each gene $z$ represented with a binary indicator $g^z \in \{0, 1\}$ to denote if any mutation occurs or not, these indicators collectively form an individual-level input vector $g \in \{0, 1\}^N$, where N is the total number of genes. To embed these discrete features into a continuous space, we treat each gene as a token and assign it a learnable embedding, yielding a sequence of gene token embeddings $[\mathbf{g}^1, \mathbf{g}^2, \ldots, \mathbf{g}^N] \in \mathbb{R}^{N \times d}$ for each patient, with $d$ denotes dimensionality.

**Aggregation Token.** Gene-level binary vectors are typically sparse and noisy, as not all genetic variants are equally associated with and contributory to disease prediction. To enable the model to adaptively focus on the most relevant subset of genes, we introduce a learnable aggregation token for each patient that attends to all gene token embeddings, aggregates information across genetic features and implicitly identifies those most predictive of the health conditions. This allows the model to dynamically prioritize informative genetic signals on a per-patient basis. Formally, we denote the learnable aggregation token by $\mathbf{v} \in \mathbb{R}^d$. Then, multi-head attention is applied with $\mathbf{v}$ as the query and the gene token embeddings as keys and values, producing a soft attention distribution over gene tokens using the standard self-attention [45]. It results in a set of non-negative mask weights

$\{m_1, \ldots, m_N\}$, where each $m_i \in (0, 1)$ and $\sum_{i=1}^{N} m_i = 1$, as determined by a softmax function. These weights quantify the relative importance of each gene in the context of the specific patient and are utilized to softly modulate the gene token embeddings. Specifically, each gene token is blended with the aggregation token following:

$$\tilde{\mathbf{g}}^i = (1 - m_i) \cdot \mathbf{g}^i + m_i \cdot \mathbf{v}, \tag{1}$$

so that highly attended genes are more strongly influenced by the aggregation token. The resulting sequence $[\mathbf{v}, \tilde{\mathbf{g}}^1, \ldots, \tilde{\mathbf{g}}^N]$ is further fed into a Transformer encoder. Ultimately, its output at the first position, corresponding to the updated aggregation token, is employed as the patient-level genetic representation $\mathbf{h}_{\mathrm{G}}$, which integrates information from all gene token embeddings to support risk prediction. The remaining contextualized outputs $\{\mathbf{h}_1, \ldots, \mathbf{h}_N\}$ from the encoder are utilized to reconstruct the original gene binary features. If features corresponding to important genes can be accurately reconstructed, it suggests that the aggregation token effectively captures and integrates individual genetic information.

**Gene Feature Reconstruction.** In addition to producing personalized genetic representations, the model is trained to reconstruct the original genetic features from the contextualized genetic outputs $\{\mathbf{h}_1, \ldots, \mathbf{h}_N\}$. This reconstruction encourages the encoder to preserve information relevant to the input genetic features, and guides the aggregation attention to concentrate on meaningful genes by penalizing reconstruction errors at attended positions. Each contextualized gene token embedding $\mathbf{h}_i$ is transformed to a binary prediction via a shared decoder as follows:

$$\hat{g}^i = \mathrm{softmax}(\mathbf{W}\mathbf{h}_i + \mathbf{b}), \tag{2}$$

where $\mathbf{W}$ and $\mathbf{b}$ are learnable parameters of the decoder. We calculate a weighted reconstruction loss using the aggregation attention scores $\{m_i\}_{i=1}^{N}$, thereby emphasizing reconstruction at highly attended positions, as defined below:

$$\mathcal{L}_{\mathrm{rec}} = \sum_{i=1}^{N} m_i \cdot \mathrm{CE}(\hat{g}^i, g^i), \tag{3}$$

where $\mathrm{CE}(\cdot)$ denotes the cross-entropy loss. It promotes effective aggregation by prioritizing informative genes and minimizing the influence of less relevant ones.

**Mixture-of-Aggregations.** Although a single aggregation token can highlight important genes, its representational capacity can be constrained when modeling complex, multifactorial biological signals. A single token is required to condense relevant gene-level information into a fixed-size vector, which might overlook the input's functional complexity, particularly for polygenic diseases where risk is distributed across multiple genetic components. To overcome this limitation and better capture the compositional patterns of genetic data, we extend the single-token design to a mixture-of-aggregations framework. Specifically, we introduce $M$ independent aggregation tokens, each functioning as a specialized expert that attends to the gene token embeddings from a distinct perspective and produces its own representation of the genetic feature. Formally, we learn a set of expert-specific aggregation tokens $\{\mathbf{v}^{(1)}, \ldots, \mathbf{v}^{(M)}; \mathbf{v}^{(k)} \in \mathbb{R}^d\}$. For each expert $k$, we compute soft attention weights over the gene token embeddings, yielding a set of mask scores $\{m_i^{(k)}\}_{i=1}^{N}$, with which expert-specific gene token embeddings are modulated:

$$\tilde{\mathbf{g}}^{i(k)} = (1 - m_i^{(k)}) \cdot \mathbf{g}^i + m_i^{(k)} \cdot \mathbf{v}^{(k)}. \tag{4}$$

The sequence $[\mathbf{v}^{(k)}, \tilde{\mathbf{g}}^{i(k)}, \ldots, \tilde{\mathbf{g}}^{N(k)}]$ is passed through the shared Transformer encoder, producing an expert-specific genetic representation $\mathbf{h}_{\mathrm{G}}^{(k)}$ and reconstruction logits $\hat{g}^{i(k)}$ for each gene using Eq. (2). To combine outputs from all experts, we adopt averaging to obtain $\mathbf{h}_{\mathrm{G}}$, $\hat{g}^i$ and $m_i$ as follows:

$$\mathbf{h}_{\mathrm{G}} = \frac{1}{M} \sum_{k=1}^{M} \mathbf{h}_{\mathrm{G}}^{(k)}, \quad \hat{g}^i = \frac{1}{M} \sum_{k=1}^{M} \hat{g}^{i(k)}, \quad m_i = \frac{1}{M} \sum_{k=1}^{M} m_i^{(k)}, \tag{5}$$

where $\hat{g}^i$ and $m_i$ are utilized in Eq. (3). This allows the reconstruction objective $\mathcal{L}_{\mathrm{rec}}$ to benefit from multi-expert knowledge. The extension encourages diverse specialization among experts and allows the model to capture multiple, potentially disentangled genetic functionalities.

## 4.2 Graph-Guided Phenotypical Representation Fine-Tuning

Beyond genetic information, EHR data as a critical resource provides rich phenotypical information for disease risk prediction. Inspired by the recent success of foundation models, we leverage the pretrained EHR foundation model MOTOR [42] to encode each patient's longitudinal medical history into a fixed-dimensional embedding. Specifically, given an EHR sequence $r$ from a patient, we obtain its embedding $\mathbf{h}_{\mathrm{E}} = \mathrm{Motor}(r)$. However, since both the source and target cohorts may differ from the data used in MOTOR pretraining, in terms of population characteristics and record conventions, the resulting embeddings generally suffer from distribution shift. Nevertheless, fine-tuning the foundation model inevitably demands substantial computational resources and labeled data, which are often unavailable in real-world local cohorts. To address this challenge, we propose a lightweight graph-guided fine-tuning strategy that adapts the pretrained embeddings to novel cohorts without updating the parameters of MOTOR itself. Instead, we leverage a graph capturing cohort-specific patterns between patients and EHR concepts to conduct fine-tuning tailored to cohort characteristics and thus optimize predictive capability.

**Graph Construction.** To facilitate the adaptation, we construct a bipartite graph $\mathcal{G} = (\mathcal{V}, \mathcal{E})$, where the node set $\mathcal{V}$ includes both patients and EHR concepts. Specifically, we define $\mathcal{V} = \mathcal{V}_p \cup \mathcal{V}_c$, where $\mathcal{V}_p = \{v_1, v_2, \ldots, v_{N_S}\}$ represents the patients in the cohort $\mathcal{S}$, and $\mathcal{V}_c = \{c_1, c_2, \ldots, c_K\}$ denotes the set of distinct medical concepts observed in the cohort. An undirected edge $(v_j, c_k) \in \mathcal{E}$ is added if patient $j$ has at least one occurrence of concept $c_k$ in their EHR record $r_j$ prior to the prediction time. The resulting graph captures co-occurrence relationships between patients and clinical events and serves as structural guidance for message passing during the adaptation process.

**Graph-guided Embedding Adaptation.** Given the bipartite graph $\mathcal{G} = (\mathcal{V}, \mathcal{E})$, we apply a graph neural network (GNN) to refine patient embeddings using local structural context. Let $\{\mathbf{h}_{\mathrm{E}}^j\}_{j=1}^{N_S}$ denote the initial MOTOR-based embeddings for the patients in $\mathcal{V}_p$, and let $\mathbf{C}^0 = \{\mathbf{c}_k\}_{k=1}^K$ denote learnable embeddings for concept nodes in $\mathcal{V}_c$. We perform $L$ layers of message passing as:

$$\mathbf{h}_{\mathrm{E}}^{j(\ell+1)} = \phi\left( \sum_{c_k \in \mathcal{N}(v_j)} \frac{1}{|\mathcal{N}(v_j)|} \cdot \mathbf{W}_c^{(\ell)} \mathbf{c}_k^{(\ell)} \right), \quad \mathbf{c}_k^{(\ell+1)} = \phi\left( \sum_{v_j \in \mathcal{N}(c_k)} \frac{1}{|\mathcal{N}(c_k)|} \cdot \mathbf{W}_v^{(\ell)} \mathbf{h}_{\mathrm{E}}^{j(\ell)} \right)$$
(6)

where $\mathbf{W}_c$ and $\mathbf{W}_v$ are learnable parameters, and $\phi(\cdot)$ is a nonlinear activation function, and $\mathcal{N}$ denotes node neighborhood. At each layer, patient nodes aggregate information from connected concept nodes, and vice versa. The resulting $\mathbf{h}_{\mathrm{E}}^{(L)}$ at layer $L$ denotes phenotypical representations for patients based on EHR information, accounting for the cohort characteristics.

## 4.3 Cross-Modal Knowledge Transfer with Distillation

To support prediction in local cohorts without genetic data, we adopt a teacher-student distillation framework that transfers knowledge from a gene-aware model (teacher) to a gene-free model (student), enabling generalization to target cohorts that lack genetic information.

The teacher model is trained on the source cohort with genetic and EHR data available. It takes as input the learned genetic representation $\mathbf{h}_{\mathrm{G}}$ from the gene encoder, along with phenotypical embedding $\mathbf{h}_{\mathrm{E}}$ after graph-guided adaptation. These inputs are concatenated and fed into the teacher model to produce output logits, as supervision for teacher training, defined as $\mathcal{L}_t$:

$$\mathbf{y}_t = f_{\mathrm{teacher}}([\mathbf{h}_{\mathrm{G}}; \mathbf{h}_{\mathrm{E}}]), \quad \mathcal{L}_{\mathrm{teacher}} = \mathrm{CE}(\mathbf{y}_t, y). \tag{7}$$

In contrast, the student model operates without access to genetic data. It relies solely on the EHR information $\mathbf{h}_{\mathrm{E}}$ to generate predictions $\mathbf{y}_s = f_{\mathrm{student}}(\mathbf{h}_{\mathrm{E}})$, where teacher and student models are implemented as multi-layer perceptrons.

Knowledge transfer is achieved through logit-level distillation, where the student is trained with a hybrid objective to integrate the supervised signal and distillation loss, which aligns its output $\mathbf{y}_s$ with the teacher's prediction $\mathbf{y}_t$. The total training loss is defined as:

$$\mathcal{L}_{\mathrm{student}} = (1 - \lambda_{\mathrm{KD}}) \cdot \mathrm{CE}(\mathbf{y}_s, y) + \lambda_{\mathrm{KD}} \cdot \mathcal{L}_{\mathrm{KD}}(\mathbf{y}_s, \mathbf{y}_t), \tag{8}$$

where $\mathcal{L}_{\text{KD}}$ is the KL divergence between student and teacher logits, CE denotes the cross-entropy loss, and $\lambda_{\text{KD}}$ is a trade-off hyperparameter. This setup allows the student to inherit decision boundaries shaped by gene-level information, thereby improving performance in gene-missing scenarios while maintaining deployment feasibility in gene-free environments.

## 4.4 Optimization

During training on the source cohort, each iteration begins by fine-tuning the representations obtained from MOTOR to produce adapted phenotypical embeddings and encoding genetic features to generate genetic embeddings. The teacher model is then trained using both the supervised loss in Eq. (7) and the reconstruction loss in Eq. (3) following $\mathcal{L}'_{\text{teacher}} = \text{CE}(\mathbf{y}_t, y) + \beta \mathcal{L}_{\text{rec}}$. The output logits from the teacher are subsequently used as supervision to train the student model via Eq. (8), with all other components kept fixed. Once training converges, the model is transferred to the target cohort for adaptation. We fine-tune the patient representation and the student model using a limited number of target patients following $\mathcal{L}'_{\text{student}} = \text{CE}(\mathbf{y}_s, y)$, applying the same graph-guided fine-tuning procedure. The fine-tuned student model is then evaluated for risk prediction on the target cohort.

# 5 Real-World Experiments

## 5.1 Experimental Setup

**Datasets.** We evaluate the proposed model $\mathbf{C}^3\mathbf{M}$ using real-world healthcare data. In this work, we focus on the Alzheimer's Disease and Related Dementias (ADRD) prediction prior to disease onset, considering its pervasive influence in the elderly population and practical data availability. We leverage the national All of Us Research Platform [39] as the source cohort, and three target cohorts respectively from one local EHR data warehouse and two sub-networks (denoted as INSIGHT-A and INSIGHT-B) from the INSIGHT Clinical Research Network [1] to simulate our setting. We identified ADRD cases and controls in these repositories and conducted preprocessing following common practices in ADRD predictive modeling [21, 41]. More dataset details can be found in Appendix A.1.

**Baselines.** As our study focuses on the realistic scenario for risk prediction, existing work fails to directly apply. Related works may target random missingness in a single domain, requiring adaptation to fit our setting, or rely on access to cross-domain data during training, which is not feasible in our scenario. To show the effectiveness of $\mathbf{C}^3\mathbf{M}$, we select two categories of methods from general domains: (1) modality imputation methods, including **CMAE** [34], **MVAE** [53], **GAN** [7] and **SMIL** [29]; (2) modality-robust learning models, consisting of **MUSE** [55], **MoMKE** [57], **DrFuse** [61] and **CMKD** [69]. For evaluation, we adapt these methods to align with our problem setup to ensure a fair comparison. Further baseline details are provided in Appendix A.2.

**Experimental Details.** We evaluate model performance using AUROC, F1 score, sensitivity, and positive predictive values (PPV). While AUROC and F1 are standard classification metrics, we additionally report sensitivity and PPV both at the 95% specificity, which are clinically relevant: the former reflects the model's ability to detect true cases under a strict false positive constraint, and the latter indicates the trustworthiness of positive predictions in clinical decision-making. We provide more detailed experimental settings in Appendix A.3.

## 5.2 Experimental Results

**Peformance Comparison.** Table 1 presents the performance of the $\mathbf{C}^3\mathbf{M}$ and baseline models on one source cohort and three target cohorts. The best results are highlighted in **bold**, while the top baseline scores are underlined. Modality imputation-based baselines, including CMAE, MVAE, GAN, and SMIL, exhibit subpar performance, likely due to their reliance on generative models to infer genetic features from EHR representations. Given the high dimensionality, noise, and sparsity in genetic data, the imputed genetic features are often inaccurate and degrade overall prediction performance. Modality robust learning models such as CMKD, MoMKE, DrFuse, and MUSE employ various strategies to integrate multimodal information, but still yield suboptimal performance. MoMKE assumes a unified encoder design across modalities; however, in our setting, EHR and genetic data exist in fundamentally different semantic spaces, making it ineffective to use a genetic encoder for processing EHR representation. CMKD attempts to align EHR and genetic embeddings, but this alignment can distort the EHR representation, reducing its informativeness and harming performance.

Table 1: Performance comparison across source and target cohorts. Metrics are in percentage (%). Sens@95 and PPV@95 represent sensitivity and PPV values at the 95% specificity, respectively.

| Method | All of Us (Source) | | | | LocalEHR (Target) | | | |
|---|---|---|---|---|---|---|---|---|
| | AUROC | F1 | Sens@95 | PPV@95 | AUROC | F1 | Sens@95 | PPV@95 |
| **CMAE** [34] | $73.09_{\pm0.55}$ | $23.36_{\pm1.66}$ | $30.26_{\pm1.74}$ | $38.66_{\pm2.30}$ | $61.22_{\pm0.36}$ | $20.72_{\pm2.16}$ | $16.75_{\pm0.91}$ | $26.17_{\pm0.30}$ |
| **MVAE** [53] | $72.35_{\pm0.49}$ | $21.79_{\pm1.85}$ | $28.07_{\pm1.66}$ | $39.03_{\pm4.23}$ | $62.06_{\pm0.10}$ | $21.04_{\pm0.78}$ | $16.57_{\pm0.47}$ | $25.93_{\pm1.31}$ |
| **GAN** [7] | $72.42_{\pm0.43}$ | $30.33_{\pm0.82}$ | $30.48_{\pm1.37}$ | $38.39_{\pm0.57}$ | $62.67_{\pm0.78}$ | $22.00_{\pm0.09}$ | $16.79_{\pm1.23}$ | $25.81_{\pm1.46}$ |
| **SMIL** [29] | $73.59_{\pm0.64}$ | $31.54_{\pm2.17}$ | $31.79_{\pm1.54}$ | $39.74_{\pm1.21}$ | $63.17_{\pm0.51}$ | $22.40_{\pm0.38}$ | $17.46_{\pm1.02}$ | $26.21_{\pm1.28}$ |
| **DrFuse** [61] | $71.95_{\pm0.71}$ | $16.02_{\pm2.96}$ | $28.07_{\pm1.37}$ | $36.46_{\pm0.66}$ | $61.47_{\pm0.73}$ | $14.00_{\pm0.80}$ | $15.85_{\pm1.05}$ | $24.97_{\pm1.49}$ |
| **MUSE** [55] | $66.81_{\pm1.72}$ | $20.97_{\pm2.91}$ | $19.50_{\pm1.96}$ | $28.98_{\pm1.49}$ | $59.23_{\pm2.10}$ | $13.13_{\pm2.58}$ | $10.52_{\pm3.72}$ | $17.93_{\pm3.19}$ |
| **CMKD** [69] | $68.34_{\pm0.94}$ | $21.55_{\pm1.38}$ | $26.39_{\pm1.97}$ | $32.30_{\pm2.90}$ | $59.47_{\pm1.03}$ | $13.67_{\pm0.60}$ | $14.39_{\pm1.54}$ | $23.81_{\pm1.68}$ |
| **MoMKE** [57] | $71.95_{\pm0.22}$ | $18.44_{\pm0.75}$ | $28.95_{\pm0.66}$ | $38.19_{\pm2.05}$ | $61.89_{\pm0.21}$ | $11.58_{\pm1.00}$ | $16.67_{\pm1.36}$ | $25.65_{\pm1.74}$ |
| **C³M** | $\mathbf{79.94}_{\pm0.34}$ | $\mathbf{36.95}_{\pm0.51}$ | $\mathbf{40.79}_{\pm0.47}$ | $\mathbf{45.93}_{\pm1.26}$ | $\mathbf{71.82}_{\pm0.31}$ | $\mathbf{31.78}_{\pm0.27}$ | $\mathbf{27.96}_{\pm1.04}$ | $\mathbf{36.66}_{\pm1.05}$ |

| Method | INSIGHT-A (Target) | | | | INSIGHT-B (Target) | | | |
|---|---|---|---|---|---|---|---|---|
| | AUROC | F1 | Sens@95 | PPV@95 | AUROC | F1 | Sens@95 | PPV@95 |
| **CMAE** [53] | $64.89_{\pm0.40}$ | $22.37_{\pm1.31}$ | $16.01_{\pm1.11}$ | $24.46_{\pm1.20}$ | $68.11_{\pm0.74}$ | $24.91_{\pm2.69}$ | $19.89_{\pm0.29}$ | $31.75_{\pm0.40}$ |
| **MVAE** [53] | $64.96_{\pm0.29}$ | $23.11_{\pm0.46}$ | $16.63_{\pm0.48}$ | $25.18_{\pm0.63}$ | $67.55_{\pm0.14}$ | $23.18_{\pm0.77}$ | $20.39_{\pm0.23}$ | $32.27_{\pm0.38}$ |
| **GAN** [7] | $65.25_{\pm0.37}$ | $22.10_{\pm2.76}$ | $16.53_{\pm0.79}$ | $25.06_{\pm0.99}$ | $68.41_{\pm0.26}$ | $25.71_{\pm1.16}$ | $20.24_{\pm0.55}$ | $32.06_{\pm0.77}$ |
| **SMIL** [29] | $65.94_{\pm0.56}$ | $22.57_{\pm1.98}$ | $17.24_{\pm0.64}$ | $25.62_{\pm1.04}$ | $69.01_{\pm0.45}$ | $26.28_{\pm1.45}$ | $20.80_{\pm0.61}$ | $32.47_{\pm0.92}$ |
| **DrFuse** [61] | $64.77_{\pm0.65}$ | $12.08_{\pm2.60}$ | $15.64_{\pm0.34}$ | $24.15_{\pm0.34}$ | $67.69_{\pm0.85}$ | $16.38_{\pm3.05}$ | $19.92_{\pm0.75}$ | $31.70_{\pm0.69}$ |
| **MUSE** [55] | $59.31_{\pm3.42}$ | $13.64_{\pm3.46}$ | $10.50_{\pm3.29}$ | $17.44_{\pm4.61}$ | $62.25_{\pm4.32}$ | $21.38_{\pm2.26}$ | $11.43_{\pm3.67}$ | $20.50_{\pm4.77}$ |
| **CMKD** [69] | $62.57_{\pm0.56}$ | $15.29_{\pm1.25}$ | $12.38_{\pm2.43}$ | $18.57_{\pm1.98}$ | $64.89_{\pm1.46}$ | $23.70_{\pm2.72}$ | $12.92_{\pm1.20}$ | $23.61_{\pm0.93}$ |
| **MoMKE** [57] | $64.77_{\pm0.40}$ | $12.92_{\pm2.06}$ | $16.28_{\pm1.08}$ | $24.75_{\pm1.21}$ | $67.38_{\pm1.11}$ | $12.44_{\pm3.57}$ | $19.56_{\pm0.61}$ | $31.36_{\pm0.78}$ |
| **C³M** | $\mathbf{72.44}_{\pm0.28}$ | $\mathbf{23.61}_{\pm1.27}$ | $\mathbf{26.63}_{\pm0.51}$ | $\mathbf{34.75}_{\pm0.62}$ | $\mathbf{75.39}_{\pm0.23}$ | $\mathbf{36.04}_{\pm1.10}$ | $\mathbf{29.68}_{\pm0.48}$ | $\mathbf{41.03}_{\pm0.63}$ |

Table 2: Ablation study across source and target cohorts.

| Method | All of Us (Source) | | | | LocalEHR (Target) | | | |
|---|---|---|---|---|---|---|---|---|
| | AUROC | F1 | Sens@95 | PPV@95 | AUROC | F1 | Sens@95 | PPV@95 |
| **C³M w/o Aggre** | $78.69_{\pm0.28}$ | $35.98_{\pm0.61}$ | $38.16_{\pm0.59}$ | $43.61_{\pm0.42}$ | $70.26_{\pm0.40}$ | $29.94_{\pm0.34}$ | $23.89_{\pm1.31}$ | $33.09_{\pm1.24}$ |
| **C³M w/o Graph** | $72.27_{\pm0.53}$ | $30.09_{\pm0.56}$ | $23.03_{\pm0.42}$ | $33.65_{\pm1.34}$ | $60.93_{\pm0.38}$ | $18.49_{\pm0.35}$ | $10.44_{\pm1.24}$ | $17.88_{\pm0.98}$ |
| **C³M w/o MOTOR** | $70.88_{\pm0.68}$ | $30.65_{\pm0.74}$ | $25.00_{\pm0.54}$ | $37.62_{\pm1.21}$ | $62.44_{\pm0.42}$ | $21.04_{\pm0.45}$ | $12.26_{\pm1.43}$ | $20.43_{\pm1.21}$ |
| **C³M w/o Gene** | $77.70_{\pm0.56}$ | $31.56_{\pm0.63}$ | $35.53_{\pm0.45}$ | $42.52_{\pm1.14}$ | $70.70_{\pm0.36}$ | $31.33_{\pm0.38}$ | $24.60_{\pm0.94}$ | $34.58_{\pm0.83}$ |
| **C³M** | $\mathbf{79.94}_{\pm0.34}$ | $\mathbf{36.95}_{\pm0.51}$ | $\mathbf{40.79}_{\pm0.47}$ | $\mathbf{45.93}_{\pm1.26}$ | $\mathbf{71.82}_{\pm0.31}$ | $\mathbf{31.78}_{\pm0.27}$ | $\mathbf{27.96}_{\pm1.04}$ | $\mathbf{36.66}_{\pm1.05}$ |

DrFuse seeks shared patterns between EHR and genetic representations, yet the inherent noise in genetic data may lead to imprecise shared representations that negatively impact downstream predictions. MUSE constructs a patient-modality graph by representing each modality as a node, which may introduce overly dense and noisy connections, further limiting its effectiveness. The proposed C³M achieves the best performance among all methods, highlighting the effectiveness of our approach and its robustness in handling phenotypical and genetic information practically.

**Ablation Study.** To assess the effectiveness of each component in our model, we conduct an ablation study on source and target cohorts, with results shown in Table 4. We evaluate four variants: (1) without the mixture-of-aggregations (w/o Aggre), (2) without the bipartite patient-concept graph for fine-tuning (w/o Graph), (3) without pre-trained EHR representations, using randomly initialized EHR representations (w/o MOTOR), and (4) without genetic data (w/o Gene). From Table 4, we observe that all ablated variants underperform the full model, highlighting the importance of each component. In particular, removing the pre-trained phenotypical EHR representation leads to a notable performance drop, demonstrating that the foundation model provides informative EHR representation and alleviates the need to train a complex EHR encoder from scratch, especially beneficial in resource-limited target sites where fine-tuning such models is challenging. The variant without the patient-concept graph shows degraded performance, suggesting that distributional shifts exist between cohorts and the pre-trained embeddings alone are insufficient to fully represent the phenotypical status of patients in new cohorts. The performance decline of the variant without the mixture-of-aggregations validates the effectiveness of our mixture-of-mask-tokens design, indicating that indiscriminately using all genes introduces noise and negatively impacts prediction. Finally, the variant without genetic features confirms the necessity of incorporating genetic information, as it contributes complementary signals that enhance phenotypical representation and risk prediction.

**Effect of the Number of Aggregation Tokens.** To explore the influence of the number of aggregation tokens, we present experimental results in Figure 3, comparing performance when varying the number of aggregation tokens from 1 to 4. In our design, aggregation tokens are used to identify informative and relevant genetic features, subsequently compressing meaningful genetic information through reconstruc-

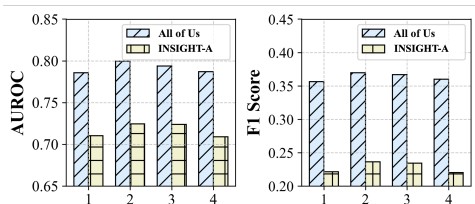

Figure 3: Effect of #aggregation tokens.

tion. The mixture-of-aggregations strategy leverages diverse perspectives from multiple experts on the same set of genes. From the results, we observe that models with 2 to 4 aggregation tokens outperform the variant using only one, likely because a single aggregation token tends to focus on a narrow subset of genes and overlook other important features. However, performance slightly degrades with 4 aggregation tokens compared to 2 or 3, possibly due to the introduction of redundant or noisy expert views that dilute the model's focus. Notably, models with 2 or 3 aggregation tokens achieve comparable performance across various metrics, indicating that a moderate number of tokens achieves a balance between diversity and relevance in capturing genetic information.

**Effect of the Number of Transformer Layers.** To investigate the impact of the number of Transformer layers, we conduct experiments with layer counts ranging from 1 to 4. As shown in Figure 4, the variant with only one Transformer layer yields inferior performance, likely due to its limited capacity to capture complex gene feature interactions and long-range dependencies. Variants with 3 and 4 layers perform

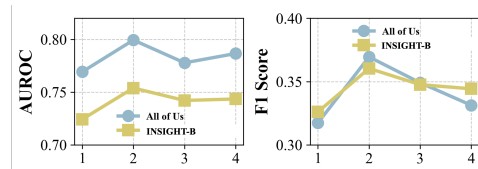

Figure 4: Effect of #transformer layers.

better but still fall short compared to the 2-layer variant, possibly because the training data size is insufficient to support effective learning in deeper models and may lead to overfitting or optimization difficulties. Notably, the 2-layer Transformer achieves the best overall performance, suggesting that it provides an optimal balance between model capacity and generalization under the current setting.

**On Alternative Graph Constructions.** In our model, we construct a patient-concept bipartite graph. An alternative approach, denoted as GCombined, integrates gene information directly into the graph by treating each gene as a concept node. To assess the impact of these two graph construction strategies, we compare their performance. As shown in Figure 5, $C^3M$ consistently outperforms GCombined across all metrics. This result suggests that incorporating genetic information directly during graph fine-tuning

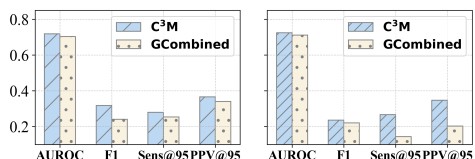

Figure 5: Performance comparison on graph construction strategies.

degrades performance, likely because the inherent noise and imprecision of genetic data negatively influence the learning of phenotypic representations derived from EHRs.

**Further Analysis.** We provide more experimental results and further analyses in Appendix B.

# 6 Conclusion

We presented a cross-cohort, cross-modal knowledge transfer framework to address the practical challenges of deploying the clinical risk prediction model in resource-limited settings. Our method enables the knowledge transfer from a multimodal model to local cohorts with only EHR data, without requiring access to genetic information during deployment. By introducing a gene encoder with mixture-of-aggregations to handle noisy genetic inputs and a lightweight graph-guided fine-tuning strategy for efficient adaptation, our framework achieved consistent and strong performance across diverse real-world cohorts. A key limitation of our current work is its focus only on Alzheimer's disease and related dementias. In future work, we plan to extend the framework to more types of chronic diseases, e.g., Parkinson's disease, to further assess its generalizability across disease domains.

## Acknowledgments and Disclosure of Funding

The authors would like to acknowledge the support from NSF 2212175, NIH RF1AG072449, RF1AG084178, R01AG080991, R01AG080624, R01AG076448, R01AG076234 and R01NS140142.

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

# A More Details on Experimental Setup

## A.1 Datasets

We evaluate the proposed method and baseline approaches with one nationwide source cohort and three local target cohorts. The nationwide All of Us Research Program is primarily utilized to build the source cohort of ADRD with EHR and genetic modalities. Three cohorts from LocalEHR, INSIGHT-A, and INSIGHT-B, are leveraged to simulate realistic, resource-limited clinical scenarios in our study.

### A.1.1 All of Us Research Program

The source cohort for ADRD prediction is assembled from the All of Us Research Program [39], launched in 2018 as a nationwide U.S. initiative designed to advance biomedical research. It collects de-identified historical health information from distributed health assessment centers standardized with the OMOP common data model [46], while retaining biological samples for detailed biological analyses. We use the All of Us dataset for its extensive, diverse cohort, which enables robust ADRD prediction across populations and heterogeneous clinical records. The cohort construction process on All of Us is illustrated as follows.

We first identify ADRD cases using clinically established criteria [21]. The onset time for each case is defined as the earlier of their first ADRD diagnosis or first anti-dementia prescription. Control samples are similarly aged individuals with no record of dementia-related diagnoses and no exposure to anti-dementia medications. To mitigate confounders and enhance clinical relevance, we construct the case-control matched cohort with a matching ratio of 1:10 following standard practices in observational health research [41] by age and clinical status. And both groups are required to be 50 years or older at onset. As neurodegenerative diseases often progress gradually, making early detection essential for timely intervention, we define the prediction time as one year prior to the disease onset, with the observation window spanning from the start of each patient's EHR records to the prediction time. Accordingly, all EHR records documented after the prediction time are excluded from model input. As a result, cases and controls are required to have at least one year of EHR history in the observation window.

Meanwhile, genetic factors have been widely implicated in the biological mechanisms underlying ADRD. These include processes such as amyloid-beta production and clearance, tau phosphorylation, lipid metabolism, and neuro-inflammatory responses [33, 31]. Additionally, genomic data Considering the accessiability of genetic data in the program, accordingly, we leverage genomic information in the source cohort, given its biological relevance to ADRD and its availability at scale, and focus on the genomic variant data, preserving alleles exceeding a population-specific frequency threshold (>1%) or allele count (>100) for a set of ADRD-related genes based on GWAS Catalog [5]. Due to partial biospecimen collection and processing constraints, genetic information is unavailable for 3,027 individuals in the All of Us cohort, resulting in a random missingness rate of 26.7%. For each individual, a gene is assigned a value of 1 in case of any mutation; otherwise, it is assigned 0. This yields binary gene features reflecting the presence of associated genetic variants, serving as the individual's genetic profile.

### A.1.2 Local Clinical Repository

The local hospital's EHR repository (LocalEHR) contains patient-level clinical information, where EHR data are standardized under the PCORnet Common Data Model [14]. Cases and controls for ADRD are defined using the same criteria as those applied in the All of Us cohort. We construct a case-control matched cohort using propensity score matching with the same ratio as the All of Us cohort based on clinical factors. Consistent criteria as All of Us for cohort construction and prediction timing are subsequently enforced for this repository.

Table 3: Dataset Statistics

| Dataset | #Patients | #Genes | Missing Rates |
|---|---|---|---|
| **All of Us** | 11,356 | 217 | 26.7% |
| **LocalEHR** | 6,548 | - | 100% |
| **INSIGHT-A** | 19,062 | - | 100% |
| **INSIGHT-B** | 17,424 | - | 100% |

### A.1.3 INSIGHT Clinical Research Network

Supported by the Patient-Centered Outcomes Research Institute (PCORI) [12], the INSIGHT Clinical Research Network (CRN) [1] aims to improve health outcomes and care delivery by analyzing healthcare utilization across major clinical and academic centers in the New York City. We use two sub-networks from the network, referred to as INSIGHT-A and INSIGHT-B, each containing longitudinal electronic health records (EHRs) spanning over ten years from diverse patient populations in the metropolitan area. All EHR data are standardized using the PCORnet Common Data Model [14]. For each sub-network, we apply identical criteria as All of Us and construct the cohort, respectively. As different clinical sites often adopt distinct data models and record protocols, the constructed cohorts at both the source and target sites are processed accordingly. In this study, we leverage demographics, diagnoses, and medications due to the major role they play in clinical decision-making. Each type of information is encoded using standard medical concept vocabularies [8, 17, 32], and the corresponding timestamps are represented as discrete events, forming a temporally ordered sequence for each patient as input to the foundation model. Data statistics are summarized in Table 3.

### A.2 Baselines

To show the effectiveness of $C^3M$, we compare $C^3M$ to eight baselines and adapt these methods to align with our problem setup to ensure a fair comparison. The baseline methods compared in our experiments include:

- **CMAE** [34] employs a cross-modal autoencoder, which is first learned on a subset of patients with complete modalities by reconstructing purposely masked-out modalities. Once trained, the model is applied to impute missing modalities for all patients. Imputation of genetic modality is performed at test time on the source cohort, and throughout on the target cohorts.

- **MVAE** [53] builds a multimodal variational autoencoder that learns a joint latent distribution robust to missing data. Leveraging a product-of-experts architecture to enable parameter sharing across modalities, MVAE takes input data with different modality combinations. We conduct MVAE for EHR and genetic modalities, and naturally skip the genetic network during inference.

- **GAN** [7] proposes a generative adversarial network for Alzheimer's disease prediction with incomplete imaging modalities. The generator reconstructs the missing modality from the available one, while the discriminator evaluates their coherence and performs classification. Inference is performed by imputing the genetic modality, followed by disease prediction using the discriminator.

- **SMIL** [29] integrates Bayesian meta-learning to jointly learn a reconstruction network, which estimates the missing modality as a weighted combination of modality-specific priors inferred from modality-complete samples, and a regularization network that conducts uncertainty-guided feature regularization. During inference, SMIL takes samples with EHR modality as input and reconstructs the genetic modality for further joint prediction.

- **MUSE** [55] represents patient–modality relationships as a bipartite graph, where patients and fine-grained EHR modalities are nodes, and modality features define the edges. A Siamese GNN is trained on original and augmented graphs to enhance robustness to missing modalities. Patient nodes are initialized with MOTOR-derived embeddings, and binary genetic features are encoded on edges connecting patients to genetic modality nodes. To ensure consistent evaluation under modality missingness, two fine-grained EHR modality nodes (diagnosis and medication) are used, with concept-level codes embedded as edge attributes. During inference, EHR-alone patients only maintain edges with fine-grained modality nodes for EHRs.

---

**Algorithm 1** The Training Procedure of $\mathbf{C^3M}$ on the Source Cohort

---

1: **Input**: Multi-modal inputs $\{(r_i, g_i, y_i)\}$, EHR foundation model Motor, bipartite graph $\mathcal{G}$;
2: **for** each iteration **do**
2:     Obtain the phenotypical representation $\mathbf{h}_E$ for each patient $i$ from Motor using $r_i$;
2:     Conduct graph-guided phenotypical representation fine-tuning and obtain $\mathbf{h}_E^{(L)}$ using Eq.(6);
2:     Encode genetic profile $g_i$, and obtain updated aggregation token as patient-level genetic representation $\mathbf{h}_G$, as well as contextualized genetic outputs using Eq.(4) and Eq.(5);
2:     Compute reconstruction loss using Eq.(2)-Eq.(5);
2:     Using phenotypical embedding $\mathbf{h}_E^{(L)}$ and genetic embedding $\mathbf{h}_G$ to optimize teacher model $f_{\text{teacher}}$ based on the loss $\mathcal{L}_{\text{teacher}}{}'$, including the reconstruction loss and classification loss;
2:     Optimize the student model $f_{\text{student}}$ using Eq.(8)
3: **end for**
3: Evaluation using the student model $f_{\text{student}}$ and trained GNN.

---

- **MoMKE** [57] adopts a two-stage learning process, consisting of unimodal expert training and experts mixing training, where a gating mechanism assigns weights based on modality availability and quality. Missing modalities are handled via zero-imputation during inference.

- **DrFuse** [61] tackles modality missingness by learning disentangled latent representations for EHR and imaging modalities. A disease-aware attention mechanism is applied to fuse shared and modality-specific representations, while selectively disregarding the network branch of a modality when it is absent. During inference, the network for genetic modality remains inactivated.

- **CMKD** [69] transfers knowledge from a stronger modality (teacher) to a weaker modality (student) with paired inputs. The student model is trained to match the intermediate representations from the teacher model and is ultimately applied with weaker modality alone during inference.

### A.3 Experimental Settings

To conduct graph-guided finetuning to obtain phenotypical representations, a two-layer GCN is adopted with 16 hidden units, along with a 16-dimensional embedding layer to represent 2634 medical concept nodes, and one transformation layer that transforms the foundation model output to initialize patient nodes. In addition, the transformer encoder consists of two layers with two heads, and we determine the expert number via search in {1,2,3,4}, while the gene decoder is an MLP with one hidden layer. Attention modulation is achieved using a multi-head attention mechanism with two heads. Both the teacher and student models are implemented as multi-layer perceptrons. The trade-off parameter $\beta$ for gene feature reconstruction is selected via grid search over {0.01, 0.05, 0.1, 0.5, 1} and set as 0.1. The trade-off parameter $\lambda_{\text{KD}}$ of knowledge distillation for the student model is set as 0.01 by grid search over {0.001, 0.005, 0.01, 0.05, 0.1, 0.5, 1.0}. The learning rate of $\mathbf{C^3M}$ and baseline models is selected from {0.01, 0.001, 0.0005}.

The proposed model is trained with the Adam optimizer employing early stopping based on validation performance. Model performance on the testing set of All of Us is reported to assess within-domain model generalizability. To simulate low-resource clinical settings, we randomly sample a small subset of each target cohort to fine-tune the model transferred from All of Us. In our main experimental results, for three target cohorts, we randomly sample 500 patients for fine-tuning, respectively. Besides, fine-tuning and evaluation on the target cohorts are both performed in CPU-only environments to assess practical deployment feasibility. Similarly, baseline approaches are trained on All of Us, and afterwards finetuned with the same small sample set on each target cohort before evaluation. Because these methods are not tailored to our scenario, we adapt their public implementations to our setting and determine hyperparameters via parameter search.

For datasets, the All of Us cohort is randomly split into training/validation/testing at a 6:2:2 ratio. The training set includes both EHR data and genetic information, with a random missingness rate of 26.7% for the genetic modality. The trained model is first evaluated on the testing set of the source cohort using EHR data alone, and further fine-tuned with a limited set of EHR-alone samples on each target cohort to assess cross-cohort generalizability. Furthermore, for a fair comparison, all baseline approaches are trained and evaluated using the same data splits and are provided with representations from the foundation model as patient features. We report the average performance as well as the

Table 4: Impact of the transferred model under varying numbers of fine-tuning patients in INSIGHT-B. "W/o T" indicates training from scratch (without using the transferred model), while "W/ T" refers to fine-tuning with the transferred model.

| #Patients | 100 | | 200 | | 300 | | 400 | | 500 | |
|---|---|---|---|---|---|---|---|---|---|---|
| | W/o T | W/ T | W/o T | W/ T | W/o T | W/ T | W/o T | W/ T | W/o T | W/ T |
| **AUROC** | 0.6620 | 0.7308 | 0.7186 | 0.7502 | 0.7241 | 0.7517 | 0.7298 | 0.7524 | 0.7421 | 0.7539 |
| **F1** | 0.2299 | 0.3522 | 0.2759 | 0.3276 | 0.2866 | 0.3541 | 0.2921 | 0.3579 | 0.3077 | 0.3604 |
| **Sens@95** | 0.1194 | 0.2803 | 0.2161 | 0.2950 | 0.2472 | 0.2961 | 0.2597 | 0.2965 | 0.2760 | 0.2968 |
| **PPV@95** | 0.2180 | 0.3955 | 0.3377 | 0.4088 | 0.3543 | 0.4080 | 0.3744 | 0.4074 | 0.3928 | 0.4103 |

standard deviation of three random runs on All of Us and the target cohorts. The training procedure of $\mathbf{C}^3\mathbf{M}$ is shown in Algorithm 1.

# B  Further Analysis and Discussion

## B.1  More Experimental Results

**Effect of the Tradeoff Parameter $\beta$.** The tradeoff parameter $\beta$ controls the balance between the supervised training objective and the gene feature reconstruction loss. We evaluate the model under different values of $\beta$ and present the results in Figure 6. A small $\beta$ leads to a weak reconstruction signal, causing suboptimal gene encoding and imperfect genetic embeddings. Conversely, a large $\beta$ shifts the optimization focus excessively toward reconstructing gene features,

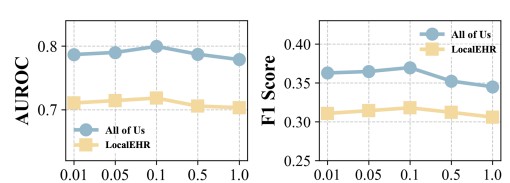

Figure 6: Effect of trade-off weight $\beta$.

which can impair the learning of task-relevant supervised representations and ultimately degrade overall performance. The best performance is achieved with a moderate value of $\beta$, indicating the importance of balancing these two objectives. Overall, the impact of the tradeoff parameter $\beta$ is not large, and $\beta$ usually slightly affects the performance.

**Effect of the Number of Patients in the Target Cohorts for Fine-tuning.** To investigate the impact of the fine-tuning dataset size, we conduct additional experiments by varying the number of labeled patients from the target cohorts used for fine-tuning. As shown in Figure 7, the model achieves quite good performance even when fine-tuned with a very small set of labeled patients. As the size of the fine-tuning set increases, the performance continues to improve, but with diminishing returns. This result highlights the robustness and practicality of our proposed

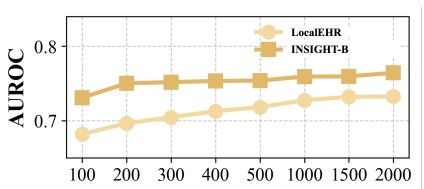

Figure 7: Effect of number of patients.

framework, particularly in resource-limited clinical settings where only a small number of labeled patients are available for fine-tuning.

**Effect of the Transferred Model.** To evaluate the effectiveness of the transferred model, we compare model performance on the target cohort (INSIGHT-B) between models trained from scratch and those fine-tuned from the transferred model's parameters. As shown in Table 4, when the number of available patients in the target cohort is limited, using the transferred model offers a significant advantage by providing a near-optimal starting point for optimization. These results underscore the necessity of studying cross-cohort knowledge transfer to enable effective model adaptation in low-resource settings.

**Time Complexity Analysis.** In our setting, target sites typically have limited computational resources, which constrain their ability to develop models independently. This motivates the need for cross-cohort knowledge transfer. To ensure our approach aligns with such constraints, we analyze the time complexity on target sites. First, the time complexity of fine-tuning the transferred model, which

consists of a GNN and student MLPs, is linear to the number of patient samples, because the time complexity of GNN is $O(d^2(|V| + |E|))$ and the time complexity of MLP is $O(d^2 \cdot N_t)$ where $d$ is the hidden dimension, $|V|$ and $|E|$ denote the number of nodes and edges in the graph, and $N_t$ is the number of patients in target cohort. Second, since the transferred model comes with pre-trained parameters that are already near the optimal solution for the target cohort, only minimal fine-tuning and a small amount of target cohort data are required. These show that our method effectively overcomes the limitations posed by resource-limited target sites.

## C   Broader impacts

This work aims to democratize clinical risk prediction by enabling the transfer of predictive models trained on multimodal nationwide data to local clinical settings with limited resources and unimodal EHR data. If widely adopted, the proposed $\mathbf{C}^3\mathbf{M}$ framework could help early risk detection, such as for Alzheimer's Disease and Related Dementias, in underserved populations and healthcare systems with constrained access to genetic testing and computational infrastructure. By reducing reliance on costly or inaccessible modalities, this approach promotes more equitable access to precision medicine tools. However, we also note the importance of local validation, continuous monitoring, and inclusive data practices in model transfer across cohorts to ensure robust and responsible model usage.

## D   More Preliminaries

**Self-Attention.**   Self-attention is a key mechanism for modeling dependencies within a sequence by computing contextualized representations for each token through interactions with all others. Given an input sequence of tokens $\mathbf{X} = \{x_1, x_2, \ldots, x_n\}$, self-attention computes the representation of each token as a weighted sum of all tokens in the sequence. This is achieved through the computation:

$$\text{Attention}(\mathbf{Q}, \mathbf{K}, \mathbf{V}) = \text{softmax}\left(\frac{\mathbf{Q}\mathbf{K}^\top}{\sqrt{d_k}}\right)\mathbf{V}, \tag{9}$$

where $\mathbf{Q}$, $\mathbf{K}$, and $\mathbf{V}$ are the query, key, and value matrices projected from the input $\mathbf{X}$, and $d_k$ is the dimensionality of the key vectors used for scaling. The attention weights capture pairwise relevance between tokens, allowing the model to dynamically focus on different parts of the sequence. Multi-head self-attention extends this mechanism by running multiple attention operations in parallel, enabling the model to capture diverse semantic relationships.

