# OpenReview forum: "Democratizing Clinical Risk Prediction with Cross-Cohort Cross-Modal Knowledge Transfer"
_NeurIPS.cc/2025/Conference — NeurIPS 2025 poster_

### Official Review · Reviewer_BXf7 · 2025-06-24

**Clarity:** 3
**Significance:** 3
**Originality:** 3
**Rating:** 4
**Confidence:** 4

**Summary:**

This paper proposes a novel cross-cohort, cross-modal knowledge transfer framework to enable effective clinical risk prediction in resource-constrained settings where only unimodal data is available. The model is trained on a multimodal source cohort with EHR and genetic data, and adapted to local target cohorts via a lightweight graph-guided fine-tuning mechanism. The framework includes a mixture-of-aggregations module to robustly encode noisy genetic features and uses teacher-student knowledge distillation for cross-modal transfer. Experiments on real-world Alzheimer’s disease datasets demonstrate improved generalization across target institutions.

**Questions:**

Please check the weakness part.

**Ethical Concerns:**

["NO or VERY MINOR ethics concerns only"]

**Final Justification:**

I think it is a useful work to address a practical problem.

But its novelty is limited.

**Limitations:**

The authors mentioned the limitations of the work and proposed future directions.

**Paper Formatting Concerns:**

No major formatting issues.

**Quality:**

3

**Strengths And Weaknesses:**

Strengths:

The paper is well-written, clearly presenting the motivation and structure of the proposed method. The key contributions of the paper include:  a strong real-world motivation and practical focus on resource-limited deployment scenarios;  novel architectural components such as the mixture-of-aggregations and graph-guided tuning;  and good reproducibility support with public code and implementation details.



Weaknesses:

The main weakness of the paper is the evaluation is narrowly focused on Alzheimer’s and related dementias. The authors propose a general framework but lack an evaluation of cross-disease generalization performance. It would strengthen the paper to include other chronic diseases to demonstrate broader applicability. Is it possible to quickly test C3M's performance on other chronic diseases?

Another major concern is that, although the paper primarily claims that the method is Cross-Cohort Cross-Modal Knowledge Transfer, the model without genetic data (w/o Gene) shows suboptimal performance in OHSU, with no significant differences from the original method on some metrics (e.g., AUROC and F1 score). Can a single target dataset effectively assess the impact of cross-modal data in knowledge transfer? In other target datasets, does the performance of the model without genetic data (w/o Gene) show a significant difference compared to the original method?

---

> ### Author Rebuttal · Authors · 2025-07-31
>
> Dear Reviewer BXf7,
>
> Thank you for the review and comments. We appreciate your recognition of the "strong real-world motivation" and the "novel architectural components". We respond to your questions below.
>
> **1. Evaluation on Alzheimer’s and related dementias.**
>
> Our focus on Alzheimer’s disease and related dementias (ADRD), including Alzheimer’s disease, vascular dementia, dementia with Lewy bodies, frontotemporal dementia, and mixed dementia, is motivated by both public health significance and technical modeling considerations. ADRD represents one of the most pressing global health challenges, currently affecting over 55 million people worldwide, with projections to triple by 2050. It is the leading cause of disability among older adults and consistently ranks among the top causes of death across countries.
>
> Unlike some chronic conditions that may present abruptly or lack well-defined preclinical stages, ADRD is characterized by a long prodromal phase, during which subtle cognitive, behavioral, and clinical changes gradually accumulate, yet often go unrecognized in routine care. This lack of definitive early biomarkers in structured EHR data further elevates the importance of computational prediction approaches.
>
> In contrast to conditions such as diabetes or hypertension, where early diagnosis is routine and evidence-based treatment guidelines are well established, ADRD remains underdiagnosed in its early stages and lacks curative treatments. This makes early detection and risk stratification not only more urgent but also more impactful for guiding clinical trials, patient monitoring, and intervention planning. In addition, ADRD is comparatively well represented in large-scale EHR datasets, offering sufficient cases for the development and validation of machine learning models, more so than some other chronic diseases with lower prevalence or more ambiguous diagnostic labeling.
>
> Lastly, due to the time-consuming nature of data preprocessing, such as defining disease cases and controls with clinical expertise, constructing appropriate cohorts, and harmonizing real-world EHRs across the four sites, we have focused our current efforts on ADRD. As noted in our discussion of future work, we plan to extend the proposed framework to other chronic neurodegenerative conditions, such as predicting Parkinson’s disease. To apply C3M to Parkinson’s disease, we will follow the same data preparation procedure described in Appendix 1.1, while incorporating domain expertise to define Parkinson’s disease cases and controls based on clinical criteria. Once the data is prepared, the C3M model can be readily applied to the Parkinson’s disease cohort without model modification.
>
>
> **2. Ablation study for genetic data**
>
> We presented ablation studies with randomly selected datasets due to space constraints in the submission. Here, we provide additional results on our model and its gene-free variant across two other target datasets: INSIGHT MS and INSIGHT Columbia in the Tables below. The results show notable differences between C3M and its gene-free variant. Besides, although the differences in AUROC and F1 metrics for OHSU are not particularly large, substantial improvements in Sensitivity@95 and PPV@95 indicate that the model performs considerably better on high-confidence predictions. This demonstrates its enhanced ability to accurately identify true positives while satisfying the stringent 95% specificity constraint, which is especially valuable in clinical decision-making contexts.
>
> | INSIGHT MS          | AUROC       | F1           | Sens@95      |   PPV@95   |
> |----------------|-------------|--------------|--------------|--------------|
> | C3M            | 72.44±0.28  | 23.61±1.27   | 26.63±0.51   | 34.75±0.62   |
> | C3M w/o Gene   | 70.26±0.34  | 21.47±1.40   | 22.17±0.47   | 31.96±0.58   |
>
> | INSIGHT Columbia          | AUROC       | F1           | Sens@95      | PPV@95     |
> |----------------|-------------|--------------|--------------|--------------|
> | C3M            | 75.39±0.23  | 36.04±1.10   | 29.68±0.48   | 41.03±0.63   |
> | C3M w/o Gene   | 73.18±0.34  | 33.86±1.32   | 24.75±0.72   | 36.52±0.59   |

---

> > ### Comment · Reviewer_BXf7 · 2025-08-06
> >
> > Dear Authors, I have read it. Thank you for addressing my questions. I have no further questions.

---

> > > ### Author Response · Authors · 2025-08-07
> > >
> > > Dear Reviewer BXf7,
> > >
> > > Thank you for recognizing our response! We're glad to hear that your concerns have been addressed.
> > >
> > > Authors

---

### Official Review · Reviewer_jjL6 · 2025-06-29

**Clarity:** 3
**Significance:** 3
**Originality:** 3
**Rating:** 5
**Confidence:** 4

**Summary:**

This paper addresses a practical challenge in developing clinical multimodal models: training on a large-scale, nationwide cohort and adapting the model to a local cohort with partially missing modalities. To tackle this, the authors propose C³M, a method designed to handle two key issues—noisy data and limited computational resources during local deployment. The approach introduces two main contributions: (1) a mixture of aggregation strategies to extract informative genetic features, and (2) a lightweight fine-tuning mechanism that adapts the model using limited patient data. Experimental results show that C³M achieves strong performance on both the source and target cohorts.

**Questions:**

1. Could the authors provide more details on missing data patterns across datasets and how these are handled during training and evaluation?
2. The results suggest that increasing the number of aggregation tokens does not yield performance gains. Is there an explanation for this saturation, and could it be task- or data-dependent?
3. Are temporal dynamics in the EHR sequences modeled or considered in any way? If not, how might this affect the method’s applicability to longitudinal clinical data?

**Ethical Concerns:**

["NO or VERY MINOR ethics concerns only"]

**Final Justification:**

My concerns are addressed. I will keep my current score of accept.

**Limitations:**

Yes

**Paper Formatting Concerns:**

No.

**Quality:**

3

**Strengths And Weaknesses:**

Strengths:
1. The paper is clearly written and well-structured, which facilitates understanding of both the motivation and technical contributions.
2. Extensive experiments are conducted on real-world datasets, with comparisons to a wide range of baselines. The inclusion of thorough sensitivity analyses further supports the model’s effectiveness.
3. The proposed approach is well-motivated for real-world deployment, addressing practical challenges such as missing modalities and resource constraints in local clinical settings.

Weaknesses:
1. The concept of cross-modal knowledge transfer via distillation is not clearly explained, and its specific contribution to performance remains unclear.
2. The paper lacks detail on the input features—both genetic and EHR—and does not provide statistics on the patient cohorts, making it difficult to assess the dataset characteristics and generalizability.
3. It is not clear how the method handles potential temporal patterns in EHR data, which are often crucial for clinical prediction tasks.
4. The code is not provided, which hinders reproducibility.

---

> ### Author Rebuttal · Authors · 2025-07-31
>
> Dear Reviewer jjL6,
>
> Thank you for the review and comments. We appreciate your recognition of the "well-motivated approach" and "clearly written and well-structured paper". And we respond to your questions below.
>
> **1. (W1) The concept of cross-modal knowledge transfer via distillation**
>
> We would like to clarify that knowledge transfer via distillation is a necessary component of our model. The teacher model, trained on the source cohort, requires both EHR and genetic data as input. Specifically, it takes phenotypical and genetic representations. However, the target cohorts contain only EHR data and lack access to genetic information, making it infeasible to directly apply the teacher model in target cohorts. To address this, we design a distillation strategy that transfers knowledge from the teacher model to a student model, which operates solely on the phenotypical representation derived from EHR data. By leveraging this distillation framework, we enable effective knowledge transfer from a gene-aware model (teacher) to a gene-free model (student), allowing the model to generalize to real-world target cohorts without genetic input. Therefore, distillation serves as a foundational component of our model; without it, the model cannot be effectively adapted to the target cohorts.
>
> **2. (W2) Details on the input features**
>
> We provide a brief introduction below, with full details in Appendix 1.1.
> - EHR features include structured, timestamped records of patient healthcare data such as diagnoses, medications, as well as demographics, standardized across sites using SNOMED and RxNorm coding vocabularies.
> - Genetic features are derived from the All of Us ACAF callset and filtered based on established SNPs recorded in the GWAS catalog associated with ADRD.
> - We describe the cohort construction, case-control matching strategy, and index date assignment in detail following the literature across all four datasets (All of Us, OHSU, INSIGHT MS, and INSIGHT Columbia), and provide the statistics on the four patient cohorts in Table 1 of the Appendix.
>
>
> **3. (W3 and Q3) How the method handles potential temporal patterns in EHR data**
>
> In our paper, we adopt the EHR foundation model MOTOR to capture temporal patterns by taking EHR sequences as input and learn patient-level embeddings, analogous to how BERT-series models are used in natural language processing to learn contextualized representations.
>
>
> **4. (W4) Reproducibility**
>
> We would like to kindly clarify that the GitHub link is included in Section 5 of the attached checklist. As the rebuttal guidelines prohibit including links in the response text, we respectfully refer the reviewer to Section 5 for access.
>
> **5. (Q1) Provide more details on missing data patterns across datasets and how these are handled during training and evaluation**
>
> Thanks for the question. We focused on a practical scenario, where the target cohorts from local healthcare providers only collect routine EHR data, while the All of Us program, as the source cohort, has measured genetic data through nationwide efforts. Therefore, the genetic data is completely missing on target datasets, namely OHSU, INSIGHT MS, and INSIGHT Columbia. Meanwhile, random missingness of genetic data also occurs on All of Us at a rate of 26.7%. Our model addresses missingness in the All of Us through the teacher–student framework, where the teacher remains fixed and the student model learns solely from EHR data when genetic information is absent.
>
>
> **6. (Q2) Explanation for the number of aggregation tokens**
>
> As shown in Figure 3 and discussed in Section 5.2, we observe that increasing the number of aggregation tokens from 1 to 2–3 improves performance, because a single token tends to focus narrowly on a subset of genetic features, potentially missing important signals. The use of multiple tokens enables a mixture-of-aggregation strategy, capturing complementary biological patterns from different perspectives.
>
> However, performance saturates or slightly declines with 4 tokens, which we attribute to the introduction of redundancy or noise. Multiple aggregation tokens may begin to extract overlapping or irrelevant information, thereby diluting the model’s discriminative capacity. This phenomenon is likely task-dependent, particularly influenced by the complexity and signal strength of the disease under study. For the task involving subtle or sparse genetic associations, increasing the number of aggregation tokens beyond three may introduce more noise than meaningful diversity, thereby limiting potential performance gains.

---

> > ### Comment · Reviewer_jjL6 · 2025-08-04
> >
> > Thank you for addressing my concerns. I will keep my current score of accept.

---

> > > ### Author Response · Authors · 2025-08-05
> > > **Thank you!**
> > >
> > > Thank you for recognizing our response! We're glad to hear that your concerns have been addressed.
> > >
> > > Authors

---

### Official Review · Reviewer_GdmP · 2025-07-03

**Clarity:** 3
**Significance:** 3
**Originality:** 3
**Rating:** 5
**Confidence:** 4

**Summary:**

This paper presents an innovative framework, C3M (Cross-Cohort Cross-Modal), designed to democratize clinical risk prediction in resource-constrained settings. The framework addresses the challenges of predicting disease risks using EHR and genetic data across different cohorts, particularly beneficial for chronic conditions like Alzheimer’s disease.

**Questions:**

1. How can the C3M framework be extended to other chronic diseases beyond ADRD, such as Parkinson’s disease or cardiovascular conditions?
2. What are the potential societal impacts and ethical considerations of deploying this framework in clinical settings? Are there any risks associated with its use that should be addressed?

By addressing these questions, the authors could further strengthen their paper and provide a more comprehensive discussion of the framework's applicability and implications.

**Ethical Concerns:**

["NO or VERY MINOR ethics concerns only"]

**Final Justification:**

Thanks to the authors for their detailed response. Ultimately the usage scenario for this model currently is established in a narrow scope - however given the possible impact I have updated my scores

**Quality:**

3

**Strengths And Weaknesses:**

There are several strong aspects about this paper
1. The paper clearly identifies a significant challenge in healthcare—predicting disease risks accurately without relying on extensive computational resources or shared data. To solve this the proposed approach (C3M) is a novel method to handle noisy genetic data and adapt EHR embeddings efficiently, making it suitable for local clinical use. The framework is well-structured with three key components, each addressing specific challenges in data handling and model adaptation.

2. Extensive experiments validate the model’s effectiveness across diverse cohorts, demonstrating its practicality and superior performance compared to existing methods.

3. The paper provides detailed descriptions of each component, enhancing understanding of how C3M operates. Further, while not explicitly detailed, the inclusion of code and experimental details supports reproducibility, which is crucial for research integrity.

Given all of these aforementioned key strengths, the paper lacks in a few aspects:
1. The framework is validated primarily on Alzheimer’s disease and related dementias (ADRD), limiting its generalizability to other chronic diseases.
2. The potential societal impacts and broader implications of the framework are not thoroughly explored.

---

> ### Author Rebuttal · Authors · 2025-07-31
>
> Dear Reviewer GdmP,
>
> Thank you for the review and comments. We appreciate your recognition of the "significant challenge in healthcare", "novel method", and "The framework is well-structured". And we respond to your questions below.
>
>
> **1. The framework is validated primarily on Alzheimer’s disease and related dementias (ADRD)**
>
> Our focus on Alzheimer’s disease and related dementias (ADRD), including Alzheimer’s disease, vascular dementia, Lewy-body dementia, frontotemporal dementia, and mixed dementia, is motivated by both public health significance and technical modeling considerations. ADRD represents one of the most pressing global health challenges, currently affecting over 55 million people worldwide, with projections to triple by 2050. It is the leading cause of disability among older adults and consistently ranks among the top causes of death across countries.
>
> Unlike some chronic conditions that may present abruptly or lack well-defined preclinical stages, ADRD is characterized by a long prodromal phase, during which subtle cognitive, behavioral, and clinical changes gradually accumulate, yet often go unrecognized in routine care. This lack of definitive early biomarkers in structured EHR data further elevates the importance of computational prediction approaches.
>
> In contrast to conditions such as diabetes or hypertension, where early diagnosis is routine and evidence-based treatment guidelines are well established, ADRD remains underdiagnosed in its early stages and lacks curative treatments. This makes early detection and risk stratification not only more urgent but also more impactful for guiding clinical trials, patient monitoring, and intervention planning.
> In addition, ADRD is comparatively well represented in large-scale EHR datasets, offering sufficient cases for the development and validation of machine learning models, more so than some other chronic diseases with lower prevalence or more ambiguous diagnostic labeling.
>
> Lastly, due to the time-consuming nature of data preprocessing, such as defining disease cases and controls with clinical expertise, constructing appropriate cohorts, and harmonizing real-world EHRs across the four sites, we have focused our current efforts on ADRD. As noted in our discussion of future work, we plan to extend the proposed framework to other chronic neurodegenerative conditions, such as Parkinson’s disease. To apply C3M to Parkinson’s disease, we will follow the same data preparation procedure described in Appendix 1.1, while incorporating domain expertise to define Parkinson’s disease cases and controls based on clinical criteria. Once the data is prepared, the C3M model can be readily applied to the Parkinson’s disease cohort without model modification.
>
>
> **2. Potential societal impacts and broader implications**
>
> This work aims to democratize clinical risk prediction by enabling the transfer of predictive models trained on multimodal, nationwide datasets to local clinical settings with limited resources and unimodal EHR data. If widely adopted, the proposed C3M framework could improve early detection and risk stratification of chronic neurodegenerative diseases, such as Alzheimer’s Disease and Related Dementias, especially in underserved populations and healthcare systems with limited access to genetic testing and computational infrastructure. By reducing reliance on costly or inaccessible modalities, this approach promotes more equitable access to precision medicine. However, careful attention is needed as performance disparities may emerge across cohorts, which is not inherent flaws of the model but practical challenges due to variations in healthcare practices. This highlights the importance of local validation, continuous monitoring, and inclusive data practices.

---

> > ### Comment · Reviewer_GdmP · 2025-08-06
> > **Thank you for your response**
> >
> > Thank you for the response.
> >
> > On the choice of Alzheimers', while I do agree with the premise on the unique degenerative and has unique pre-diagnosis stages, it brings into the point of the large-scale applicability of the method.
> >
> > Moving beyond this point, given the very goal of democratizing the models, there are several key apsects missing currently.
> > For example, how do you propose to control the usage in a  manner consistent with the training regime of the model? Given the models could be used in many different settings, detailed post-hoc model evaluations across meaningful sub-groups, robustness tests, and fairness evaluations are missing. Have you conducted these tests?

---

> > > ### Author Response · Authors · 2025-08-07
> > >
> > > Thank you for your feedback and questions. We respond to your questions below.
> > >
> > > **1. How do you propose to control the usage in a manner consistent with the training regime of the model?**
> > >
> > > We would like to clarify that the deployment of our model at target sites is easily manageable because we intentionally designed it as a pre-training and fine-tuning pipeline, which can ensure the consistent utility of training. For the target cohorts, only fine-tuning is required. Specifically, the use of the pretrained MOTOR model allows target sites to efficiently derive patient representations with an interoperable pipeline. Additionally, the graph-guided phenotypical representation fine-tuning relies solely on local data. This whole pretraining and fine-tuning strategy avoids complex designs that could impede rapid adoption and deployment. As discussed in the appendix, the fine-tuning time on the OHSU and INSIGHT-MS cohorts using consumer-grade CPUs is approximately 300 and 560 seconds, respectively, substantially more efficient than training a model from scratch, further demonstrating the ease of adoption.
> > >
> > > **2. Robustness**
> > >
> > > Regarding robustness, we would like to emphasize that our evaluation was conducted on four cohorts: All of Us, OHSU EHR data warehouse, INSIGHT-MS and INSIGHT-Columbia from the INSIGHT Clinical Network. The All of Us program recruits participants all over the United States. In contrast, INSIGHT-MS and INSIGHT-Columbia represent different patient populations from the New York City metropolitan area, while the OHSU EHR data warehouse reflects a local patient population from Oregon. The different demographic composition of patient populations in these different evaluation data sets can ensure the robustness of the model performance.
> > >
> > > As shown in Table 1 of the paper, performances across three target cohorts (OHSU, INSIGHT-MS, and INSIGHT-Columbia) are relatively consistent, with performance variations mainly attributable to differences in cohort-specific characteristics. The All of Us yields higher performances due to the fact that it serves as the source cohort and is therefore not subject to distribution shift. In addition, the ablation study in Table 2 shows that performance on the target cohorts drops significantly when key design components are removed, underscoring the importance of our holistic model architecture. Moreover, Figure 2 in the Appendix presents additional results illustrating the impact of the number of patients in the target cohort used for fine-tuning. Performance remains stable when the cohort size exceeds 200, suggesting that our framework is robust and practical. Overall, the consistent performance across diverse target cohorts with different experimental settings demonstrates the robustness of the proposed method.
> > >
> > > **3. Sub-groups and fairness**
> > >
> > > Regarding the post-hoc model evaluations, we provide additional experimental results below.
> > >
> > > |    | AUROC (All of Us)  | F1 (All of Us) |  AUROC (INSIGHT MS)|  F1 (INSIGHT MS)|
> > > | -------- | ------- | ------- | ------- | ------- |
> > > | Female  |  79.46   |  36.57  |   72.27  |   23.41   |
> > > | Male |   80.22   |  37.03  |  72.59   |  23.84    |
> > > | -------- | ------- | ------- | ------- | ------- |
> > > | White    |  78.60   |   36.84   |   71.60  |   23.35   |
> > > | Black    |  77.42   |   36.62   |  70.92   |    22.89  |
> > > | Others    |  81.32   |  38.24   |  73.71   |  24.85    |
> > > | -------- | ------- | ------- | ------- | ------- |
> > > | Hispanic    |   81.58  |   38.54    |  72.84   |  23.92    |
> > > | Non-Hispanic    |   78.81  |   36.39    |  72.17   |   23.41   |
> > >
> > > We consider subgroups defined by sex (female and male), race (White, Black, and others), and ethnicity (Hispanic and non-Hispanic individuals). The performance across these subgroup comparisons is relatively on par, indicating that the proposed model maintains consistent accuracy and fairness across diverse demographic groups.
> > >
> > > In addition, we emphasize that our setting is designed to transfer a pre-trained model from a multimodal source site with sufficient computational capacity to unimodal target sites with limited resources. This promotes fairness from a resource-access perspective, enabling accurate risk prediction in local cohorts that lack advanced infrastructure and rely solely on EHR data.
> > >
> > > We acknowledge the importance of ensuring fairness in developing AI algorithms for medicine. Our motivation to enable knowledge transfer and alleviate resource and modality limitations aligns with the direction of equitable model utility. We believe our framework contributes to improving accessibility in resource-limited settings, and we hope it can motivate future studies that explore fairness more comprehensively in this context.
> > >
> > >
> > > Please let us know if you have further questions or concerns. Thank you!

---

### Official Review · Reviewer_aRk9 · 2025-07-03

**Clarity:** 3
**Significance:** 2
**Originality:** 2
**Rating:** 4
**Confidence:** 4

**Summary:**

This paper focuses on a specific setting for clinical risk prediction: transferring predictive models across cohorts that differ in data availability, specifically from a national cohort with access to both genetic and electronic health record (EHR) data to resource-limited cohorts with only EHR data. To tackle this, the authors propose a cross-cohort, cross-modal knowledge transfer framework, $\mathbf{C^3M}$, which integrates a mixture-of-aggregations design with a lightweight, graph-guided fine-tuning strategy. The proposed $\mathbf{C^3M}$ is evaluated on real-world clinical datasets to assess its effectiveness.

**Questions:**

Please refer to points W1 through W7 above for detailed concerns and specific recommendations related to the problem setting, technical novelty, and experimental evaluation, among other aspects of the paper.

**Ethical Concerns:**

["NO or VERY MINOR ethics concerns only"]

**Final Justification:**

I appreciate the authors' comprehensive rebuttal, which addresses most of my earlier concerns—particularly the newly added discussions on practical relevance and general applicability, the expanded ablation experiments, and the efficiency analyses. In light of these, I am pleased to increase my rating.

**Limitations:**

Yes

**Quality:**

3

**Strengths And Weaknesses:**

**S1.** The problem of clinical risk prediction, especially in cross-cohort and cross-modality scenarios, is of clear clinical and scientific importance. Addressing this challenge requires the development of technically sound and transferable methodologies to improve predictive performance across heterogeneous data sources.

**S2.** The proposed $\mathbf{C^3M}$ framework consists of three main components: Gene Encoding with Mixture-of-Aggregations, Graph-Guided Phenotypical Representation Fine-Tuning, and Cross-Modal Knowledge Transfer via Distillation. The overall design is reasonable from a technical standpoint.

**S3.** The experimental evaluation includes comparisons with relevant baselines, ablation studies to assess the contribution of each component, sensitivity analyses on key hyperparameters, and computational efficiency assessments.

**S4.** The paper is clearly written and well-structured, which supports readability and facilitates understanding of the technical contributions.

**W1.** The problem setting investigated in this work is relatively narrow, as it assumes the availability of both genetic and EHR modalities in a global (source) cohort, and the need to transfer the learned clinical risk prediction model to target cohorts with only EHR data. While this setting is technically well-defined, its practical relevance and general applicability are not sufficiently justified. Moreover, in the related work discussion, the authors note that existing methods are not directly comparable due to differences in problem settings. This further underscores the need for a more detailed explanation of the real-world motivation behind the proposed setting. Specifically, the authors had better elaborate on why this transfer setting is more realistic, e.g., how often such scenarios arise in practice.

**W2.** A notable concern pertains to the limited technical novelty of the proposed framework. While $\mathbf{C^3M}$ integrates several components—such as Mixture-of-Aggregations, graph-guided fine-tuning, and knowledge distillation—these are primarily based on off-the-shelf techniques, and the paper does not clearly articulate the specific technical challenges encountered or addressed in combining them. Furthermore, the two key challenges identified in the introduction as motivations for $\mathbf{C^3M}$ have been studied in prior work. As a result, the contributions of the paper may appear incremental, and the originality of the framework is not convincingly established.

**W3.** The experimental evaluation exhibits inconsistency in dataset selection across different analyses. For example, the ablation study is conducted on All of Us and OHSU. Figure 3 reports results on All of Us and INSIGHT MS, while Figure 4 involves All of Us and INSIGHT Columbia. This inconsistent usage of datasets without a clear justification weakens the coherence and comparability of the experimental findings. To enhance the credibility of the evaluation, it would be important either to standardize the dataset choices across experiments or to provide a clear rationale for the selection of specific dataset pairs in each analysis.

**W4.** The paper would benefit from the inclusion of case studies or qualitative analyses that provide interpretable insights into the learned representations. For example, in the Mixture-of-Aggregations component, it would be informative to demonstrate how the model encourages diverse specialization among experts through the proposed extension, potentially capturing distinct and disentangled aspects of genetic functionality. Such analyses would not only enhance interpretability but also help concretely illustrate the clinical relevance and advantages of $\mathbf{C^3M}$ in supporting clinical risk prediction.

**W5.** Given the focus on clinical risk prediction—a domain with significant clinical implications—a medical validation would be valuable to assess the practical utility of the proposed $\mathbf{C^3M}$ framework in real-world medical decision-making.

**W6.** According to the dataset statistics in Table 1 (appendix), the global cohort includes more patients than the local cohorts. This is somewhat counterintuitive and would benefit from clarification regarding cohort definitions and the rationale behind this setup.

**W7.** In the computation analysis, it would be helpful to include corresponding results for the baseline methods to better contextualize how the proposed framework balances predictive performance and computational efficiency relative to existing approaches.

---

> ### Author Rebuttal · Authors · 2025-07-31
>
> Dear Reviewer aRk9,
>
> Thank you for the review and comments. We respond to your questions below.
>
> **W1. The practical relevance and general applicability of the setting are not sufficiently justified**
>
> We respectfully argue that our work utilizes a source cohort from the All of Us Program, a large-scale biomedical initiative launched in 2015 by the National Institutes of Health. As one of the most comprehensive and demographically diverse biobanks, it represents a nationwide effort and remains openly accessible to researchers worldwide, enabling studies on various health conditions [1]. Our problem setting reflects a growing trend in translational biomedical AI [2], and aligns with current priorities of the HHS Plan 2025 [3], which calls for democratizing AI to reduce the “digital divide” for underserved populations and under-resourced healthcare systems, where models must bridge the gap between rich-resource training environments and real-world clinical deployment.
>
> However, compared with well-funded biobanks, most clinical institutions (e.g., community hospitals, regional health systems) only retain structured EHRs in routine practice, lacking genetic testing due to cost, infrastructure, or policy constraints. This results in a practical distribution shift between well-resourced source cohorts with both EHR and genetic data, and target scenarios with EHRs only.
>
> Our goal is to capitalize on biomedical signals available in multimodal nationwide cohorts (e.g., All of Us) to improve clinical risk prediction in resource-limited settings. This transfer setting is particularly realistic in ongoing nationwide efforts to democratize precision medicine [3], where insights derived from well-characterized research cohorts should generalize to more typical clinical settings. The target real-world clinical repositories,  e.g., OHSU and INSIGHT, only contain EHRs without accompanying genetic data. Our work explicitly addresses this critical yet underexplored setting, which we believe will grow increasingly relevant as genomics-informed models gain traction, but data gaps remain across clinical institutions.
>
> [1] All of Us Research Program Investigators. The 'All of Us' research program. New England Journal of Medicine. 2019.
>
> [2] Biobanking with genetics shapes precision medicine and global health, Nature Reviews Genetics, 2024.
>
> [3] U.S. Department of Health and Human Services. Strategic Plan for the Use of Artificial Intelligence in Health, Human Services, and Public Health. 2025.
>
> **W2. Limited technical novelty**
>
> We respectfully clarify that our paper addresses a critical yet underexplored task, cross-cohort cross-modal knowledge transfer, and proposes a framework specifically designed for this setting, which, to our knowledge, has not been studied before. While our method incorporates concepts such as graph learning and knowledge distillation, these techniques are not simply and directly applied in our setting. Instead, we introduce novel designs specifically tailored to the unique challenges of this problem. We emphasize that our goal is not to combine existing concepts simply, but to design a cohesive system where each element is purposefully constructed to serve the overarching objective, rather than being used as standalone techniques.
>
> In particular, our mixture-of-aggregation module for genetic data is fundamentally different from the standard mixture-of-experts approach, which routes input to relevant neural networks for training. In our design, we introduce the aggregation token to capture patient-level genetic information, accounting for the noise and context dependency in genetic expression; meanwhile, we leverage multiple tokens to model the cumulative effects across different genetic pathways, enabling fine-grained fusion on modality-specific signals. Likewise, the graph-guided fine-tuning strategy also diverges from conventional graph learning or traditional fine-tuning. Rather than updating large pretrained models, we adopt a lightweight, graph-based mechanism to adapt patient representations by leveraging cohort-specific healthcare patterns. This enhances adaptability to population, coding convention, and disease pattern shifts across sites, while preserving efficiency in low-resource settings. Importantly, knowledge transfer is a necessary part of our solution, serving the ultimate goal of transferring knowledge from rich-resourced cohorts to the target ones. To our best knowledge, both the mixture-of-aggregation and the graph-guided phenotypical representation fine-tuning components are novel contributions, thoughtfully developed to meet the particular demands of cross-cohort cross-modal clinical risk prediction.
>
> We would also like to note that, rather than being purely motivated by methodological challenges, our work is grounded in a real-world problem: the limited availability of high-cost data modalities and limited computational resources in many regional healthcare systems. While issues such as noisy or imprecise multimodal data have been studied in domains like CV and NLP, they remain underexplored, particularly in the context of genetic and EHR data fusion. Canonical multimodal methods are not capable of handling our setting, as discussed in related work. Moreover, the challenge of limited computation at local cohorts is a practical constraint to our cross-cohort setting. This particular need and task setup remain unaddressed, underscoring the novelty and significance of our contribution.
>
> **W3. Difference in used datasets**
>
> We would like to clarify that the dataset for the ablation study was randomly chosen, considering space limit and figure readability, and not guided by any specific selection criteria. We provide additional experimental results on the OHSU cohort below to augment the analyses presented in Figures 3 and 4.
>
> |Figure 3 |1|2|3|4 |
> |--------|--------|--------|--------|--------|
> |AUROC |70.35|71.82|71.76|70.53|
> |F1|30.24|31.78|31.59|30.64|
>
> | Figure 4 |1|2|3|4|
> |--------|--------|--------|--------|--------|
> | AUROC| 68.71| 71.82|70.68|70.36|
> | F1| 27.82|31.78|29.41|29.27|
>
>
> For Figures 3 and 4, the additional results from the OHSU dataset show a consistent pattern with those observed in the All of Us, INSIGHT-MS cohorts, and INSIGHT-Columbia. We will standardize the dataset choices in revision.
>
> **W4. Case studies or qualitative analyses**
>
> Thank you for the suggestion. We will include t-SNE visualizations in the revision as case studies to illustrate how gene representations are distributed under different aggregation tokens before being passed to the Transformer encoder. While t-SNE provides an intuitive view of representational differences, we are unable to include additional figures during the rebuttal phase. As a quantitative alternative, we compute the Earth Mover’s Distance (EMD) between the two clusters of gene embeddings when using two aggregation tokens. EMD measures the minimal effort required to transform one distribution into another, making it suitable for comparing embedding distributions. In our case, the EMD is approximately 5.6, versus 0 for identical distributions. This non-zero value indicates that the representations learned from the two aggregation tokens are indeed distinct, supporting the hypothesis that our design promotes diverse and potentially disentangled genetic representations.
>
> **W5. Medical validation**
>
> We respectfully note that our validation on the OHSU and INSIGHT cohorts can be viewed as a form of practical medical validation. These target EHR datasets are derived from real-world hospitals actively serving patients, containing routinely collected records from genuine clinical encounters in Oregon and New York City. Our cohort construction adheres closely to domain expertise and established medical research criteria. As such, our target cohort results demonstrate the model’s practical utility and robustness in realistic clinical environments. We also fully agree that direct medical validation is an important next step and plan to collaborate with our clinical partners to assess the alignment between model outputs and clinical decision-making processes.
>
>
> **W6. Dataset statistics**
>
> Though we utilized the “source cohort” term in the paper, it has no implication on the patient number but refers to a nationwide population and biological modality availability. The All of Us program recruits participants nationwide, forming one of the most diverse health research datasets to date. In contrast, both INSIGHT-MS and INSIGHT-Columbia represent patient populations across the NYC area, which enables larger patient coverage due to the region's high population density.
>
> The patients in each cohort are identified through a strict procedure with clinically relevant definitions and filtering criteria; the final cohort size depends only on the covered population scale and the cohort construction procedure, for which we kindly refer the reviewer to the cohort descriptions in Appendix 1.1.
>
> Importantly, the work focuses not on absolute cohort sizes, but rather on addressing real-world disparities between cohorts, from a national biobank with diverse, representative populations to regional populations and systems; from multimodal data (EHR + genetics) to unimodal settings (EHR only); and from high-resource computing environments to resource-constrained clinical sites. This setup simulates the practical challenges in translating risk prediction models into more typical real-world healthcare settings.
>
> **W7. Computational efficiency of baselines**
>
> We provide a comparison between our model and two representative baselines, SMIL and MoMKE. The fine-tuning time on the OHSU is approximately 300s for our method (Appendix 2.1 (Computation Analysis)). In contrast, SMIL and MoMKE require approximately 2× and 4.3× longer, respectively, under the same computing setup. Our method shows significantly lower computational overhead and better predictive performance.

---

> > ### Comment · Reviewer_aRk9 · 2025-08-05
> >
> > I appreciate the authors' comprehensive rebuttal, which addresses most of my earlier concerns—particularly the newly added discussions on practical relevance and general applicability, the expanded ablation experiments, and the efficiency analyses. In light of these, I am pleased to increase my rating.

---

> > > ### Author Response · Authors · 2025-08-05
> > > **Thank you!**
> > >
> > > Thank you for recognizing our response! We're glad to hear that your concerns have been resolved.
> > >
> > > Authors

---

### Author Response · Authors · 2025-08-09
**General Response**

Dear Reviewers and ACs,

As the discussion nears its conclusion, we sincerely thank all reviewers for their insightful comments and suggestions, and we appreciate your recognition of our work. As noted, our work identifies and addresses a significant and practical challenge in healthcare (Reviewer GdmP, jjL6, BXf7) and proposes a reasonable (Reviewer aRk9), novel, and well-structured framework (Reviewer GdmP, BXf7) that is well-motivated for real-world deployment (Reviewer jjL6, BXf7). The paper is clearly written (Reviewer aRk9, jjL6, BXf7) with good reproducibility support (Reviewer BXf7, GdmP).

We greatly appreciate your constructive feedback and recognition of our responses. And we hope the discussion has adequately addressed any concerns regarding this study.

Thank you once again for your time and effort in reviewing our work.

Best regards,

Authors

---

### Decision · Program_Chairs · 2025-09-17

**Decision:**

Accept (poster)

**Comment:**

After carefully evaluating the paper and considering the reviewers' feedback, we recommend acceptance of this work. The paper presents a well-motivated and technically sound framework, C3M, for cross-cohort, cross-modal clinical risk prediction, addressing an important challenge in healthcare AI: transferring models from resource-rich to resource-limited settings. Despite minor limitations, this work makes a valuable contribution to clinical AI by enabling robust cross-modal knowledge transfer in resource-constrained settings. The methodological rigor, clear presentation, and practical relevance justify acceptance. We encourage the authors to address the reviewers' concerns—particularly regarding generalizability and dataset consistency—in the final version.